# Research on Evolutionary Game of Water Environment Governance Behavior from the Perspective of Public Participation

**DOI:** 10.3390/ijerph192214732

**Published:** 2022-11-09

**Authors:** Meng Sun, Xukuo Gao, Jinze Li, Xiaodong Jing

**Affiliations:** 1School of Management, Xi’an University of Architecture and Technology, Xi’an 710055, China; 2Business School, Hohai University, Nanjing 211100, China

**Keywords:** public participation, collaborative governance, evolutionary game

## Abstract

As an informal environmental regulation, public participation plays a vital role in the multi-governance environmental system. Based on the evolutionary game theory, this paper constructs the game models of government enterprise, public enterprise and government public enterprise, and analyzes the impact of different intensity of government behavior and public participation on enterprise behavior strategies. The results show that: (1) In the two-party evolutionary game, the behavior of each stakeholder is related to its costs and benefits. Still, effective public participation allows the enterprise to choose legal discharge, even if the benefits of legal discharge are smaller than illegal discharge. (2) In the three-party evolutionary game, the steady-state conditions of government and the public are the same as those in two-party evolutionary game models. However, the decision-making behavior of enterprises also needed to consider the impact of public whistle-blowing on their reputation and image. (3) With the increase of the government’s ecological protection publicity, subsidies, fines, public concern, and whistle-blowing, the evolution speed of the enterprise towards legal discharge is faster.

## 1. Introduction

Water is not only a basic natural resource, but also a strategic economic resource related to the national economy and the people’s livelihood. Water security is related to national economic and social development and stability, as well as human health and well-being [1]. In recent years, with the intensification of the contradiction between China’s social economy and the shortage of water resources, the problem of water resource pollution has gradually attracted the public’s attention [2,3]. As a country suffering from severe drought and water shortage, China’s per capital water resources are only 1/4 of the world average and 1/5 of the United States [4]. To effectively alleviate water poverty and improve water resource utilization efficiency, the Chinese government has taken water environment governance as one of the most critical environmental problems and given close attention and policy support to water environment restoration. In 2022, the Ministry of Water Resources, the Ministry of Finance, the National Development and Reform Commission, and other departments jointly released the “Industrial Water Efficiency Improvement Action Plan,” which pointed out improving the industrial water-saving policy mechanism and enhancing the enterprise’s water-saving awareness. The outline of China’s “the 14th Five-Year Plan (2021–2025)” also clearly points out that it is necessary to implement the deadline to discharge industrial pollution sources and improve the coordinated mechanism of water pollution prevention and control in river basins. In practice, China has been committed to environmental protection and pollution control. However, some enterprises still have illegal behaviors such as illegal discharge, leakage, and excessive discharge. In 2020, enterprises’ vicious pollution discharge events caused severe damage to ecological environment safety and people’s health, such as the “Heilongjiang tailings pond leakage [5].” Therefore, effectively curbing enterprises’ pollution discharge and exploring reasonable and practical environmental policies have become essential to ensure water resources’ safety and promote social and economic development.

Water environment governance has been the focus of global scholars [6,7,8]. However, most studies explore the water environment governance from the macro policy perspective. As the policymaker of environmental regulation, the government often takes measures such as environmental protection tax [9], investment in pollution control [10], and environmental subsidies [11] to restrict the emission behavior of enterprises. However, water environment governance issues involve multiple stakeholders, and different stakeholders have their interest demands [12]. On the one hand, under the performance appraisal system based on economic growth, strict environmental regulations will cause the decline of the local economy, which is not conducive to the promotion of local government employees [13]. At the same time, stringent environmental regulation will cause higher law enforcement costs. Therefore, the government often provides poor regulation and even allows the enterprise to discharge illegally [14]. On the other hand, environmental regulation aggravates operating costs, leading to a decline in enterprise performance and market competitiveness. As an economically rational person, enterprises are motivated to discharge illegally without strict supervision. Therefore, water environment governance is a systematic project with integrity and complexity characteristics, which is difficult to achieve an ideal governance effect only by relying on the participation of government and enterprises. In recent years, with the continuous improvement of public environmental awareness, more and more people have begun to exercise the authority to supervise the behavior of the government and enterprises through government hotlines or media, and other channels [15]. As an informal environmental regulation, public participation can effectively compensate for the shortcomings of “government intervention” and “market mechanism” to provide empirical evidence for diversified ecological governance [16].

Pollution control involves multiple stakeholders, and it is challenging to control enterprise pollution discharge effectively only by relying on government regulation [17]. The essence of environmental governance is cooperation based on market principles, public interests, and legitimacy [18]. Its management system not only depends on government authority but also should fully play the critical role of public participation. Public participation is an effective measure to restrain the discharge behavior of enterprises and urge the government to strictly supervise environmental governance. Sun [19] considers the public’s environmental awareness and analyzes the enterprise’s optimal strategy under different competitive behaviors and policy contexts. When the public perceives that the government lacks adequate supervision over enterprises, they will pressure the government through complaints, petitions, and other ways. Many studies have demonstrated the importance of public participation in environmental governance from the aspects of participatory mechanism [20,21], participation mode [22,23,24,25], public attention [26,27,28], and so on. In 2008, “Measures for the publicity of Environmental Information (for trial implementation)” further provided the basis for public participation in environmental protection from the legal level. Especially under the national strategic deployment of promoting ecological civilization construction, diversified participation and social co-governance have become the water environment governance theory and practice trend.

The problem of water environment governance not only stays at the technical level but also is a practical dilemma under different stakeholders’ interest conflicts. Evolutionary game theory based on bounded rationality and group behavior analysis is increasingly used to reveal the behavior of complex subjects in environmental governance [15,19,29,30]. The existing studies provide a reference for the governance path of the water environment, but there are still the following deficiencies. On the one hand, the governance effect is often affected by the government, enterprises, the public and other stakeholders, while most studies only focus on the role of the government or the public and enterprises in the water environment governance system. On the other hand, many studies have discussed the influence of stakeholders on enterprise decision-making but have not yet considered the implementation strength of different stakeholders. However, multiple forms of government behavior and public participation often affect water environment governance. As a low-cost, efficient, and sustainable approach to water environment management, the public participation system is usually applied in practice in isolation from government governance. In addition, different forms of government behavior and public participation will affect the choice strategies of enterprises. Thus, this paper constructs an evolutionary game model of government, enterprises, and the public based on the bounded rationality assumption, considering the implementation strength of government and the public in water environment management. Then, through numerical simulation, this paper analyzes the impact of different intensities of government behavior and public participation on the choice of corporate behavior strategies, hoping to provide policy recommendations for building an environmental governance system dominated by the government, enterprises, and the public.

The remainder of this paper is structured as follows: Section 2 describes the problem and constructs two-party and three-party evolutionary game models. Section 3 reveals evolutionarily stable strategies between different stakeholders through numerical simulation analysis. Section 4 is the experimental discussion. Section 5 presents the conclusions and discusses future research directions.

## 2. Materials and Methods

Due to the complexity of the water environment governance system, water pollution control is not only the responsibility of a particular subject but also requires cooperation among the government, enterprises, the public, and other relevant stakeholders. Thus, this paper constructs the two-party evolutionary game model of government and enterprises, the public and enterprises, and the three-party evolutionary game model of government, enterprises, and the public to explore the evolutionary stable strategies (ESSs) of different stakeholders.

The complexity of social contexts and decision-making problems makes the traditional game model based on complete rationality challenging to implement in reality. On the contrary, the evolutionary game model, which is based on bounded rationality and analyzes the behavior of each stakeholder to achieve an equilibrium state in a dynamic framework, has been widely used in the environment field [31,32]. Therefore, this paper assumes that each subject has two behavior strategies after multiple games. The government regulates other subjects actively to participate in water environment governance through supervision, fines, or subsidies, and the strategy choice is recorded as positive regulation or negative regulation. Enterprises strengthen their law-abiding awareness and implement legal responsibility for environmental protection, and their strategy choice is recorded as legal discharge or illegal discharge. The strategic choice of the public is recorded as active participation or non-participation in environmental governance. Based on the above analysis, this paper sets out the following hypotheses:

**Hypothesis** **1.***In the “natural” environment, without considering other constraints, the system composed of government, enterprises, and the public is regarded as a complete system. The three parties in the system are bounded rationally in the game process and reach the equilibrium state through multiple games*. x, y*, and*z*, respectively, represent the probability of positive regulation of government, legal discharge of enterprises, and active participation of the public, all of which are functions of time*t, *satisfying*0≤x(t)≤1, 0≤y(t)≤1, 0≤z(t)≤1.

**Hypothesis** **2.***As for the government, its behavior is divided into two categories in environmental governance: incentives and punishments. The former includes the costs of promotion ecological protection publicity and the incentive costs when the enterprise meets the subsidy criteria. These are recorded as*αA*and*βB, *respectively, where*α*and*β*are the implementation intensity factors. The latter refers to the government fines when the enterprise chooses the illegal discharge and are recorded as*λD*, with*λ*being its implementation intensity factors. The social benefits, such as improving the local environment and promoting regional economic development, that the government receives when the enterprise chooses legal discharge are recorded as*$R_{1}$*. When an enterprise discharges illegally, the government gains losses, such as inhibiting the process of regional economic development, which are recorded as*$C_{1}$*. In addition, the losses of government credibility caused by the public reporting are recorded as*$C_{2}$.

**Hypothesis** **3.***As for enterprises, when they discharge legally, the economic benefits are recorded as*$R_{2}$*and the additional benefits, such as the improvement of enterprises’ reputation and image, are recorded as*$R_{3}$*. When the enterprise discharges illegally, the economic benefits are recorded as*$R_{4}$*. Additionally, the losses are recorded as*$C_{3}$*, such as the loss of enterprises’ reputation and image. The costs of legal and illegal discharge are recorded as*$C_{4}$*and*$C_{6}$*, respectively (*$C_{4}>C_{6}$*)*.

**Hypothesis** **4.***As for the public, its behavior is divided into two categories in environmental governance: public concern and public reporting, and its costs are recorded as*εF*and*σG*, respectively, where*ε*and*σ*are the implementation intensity factors. The benefits to the public are recorded as*$R_{5}$*when an enterprise legally discharges its pollutants. The losses to the public are recorded as*$C_{5}$*when the enterprise discharges illegally. In addition, the additional benefits to the public from participation are recorded as*$R_{6}$.

The above parameters and descriptions are shown in Table 1.

## 3. Analysis of Evolutionary Stability Strategies among Different Stakeholders

### 3.1. Evolutionary Game Model of Different Subjects

#### 3.1.1. The Equilibrium Analysis of Government and Enterprises

This paper divides the government’s behavior into two categories in water environmental governance: incentives and punishments, and further introduces environmental protection publicity and technological innovation incentives into incentive behavior. On the one hand, the government’s publicity is conducive to raising public awareness of environmental protection. Additionally, effective public participation can compensate for the shortcomings of government administration, indirectly promoting the transformation and upgradation of enterprises, the management of water environment, and the consequent effectiveness of environmental management. On the other hand, the government’s preferential policies such as subsidies, tax reduction, etc. for enterprises legal discharge can be regarded as an incentive means of direct support. In addition, as one of the administrative punishment methods, fines can effectively curb enterprises’ violations of environmental regulations. This paper constructs the payoff matrix of government and enterprises, as shown in Table 2.

The expected and average return of the government’s positive regulation and passive regulation can be expressed as follows:(1)$$u_{a}=y\left(R_{1}-\alpha A-\beta B\right)+(1-y)(\lambda D-\alpha A-C_{1})$$
(2)$$u_{b}=y\left(R_{1}\right)+(1-y)(-C_{1})$$
(3)$$\bar{u_{1}}=xu_{a}+(1-x)u_{b}$$

The expected and average return of the enterprises’ legal discharge and illegal discharge can be expressed as follows:(4)$$u_{c}=x(R_{2}+R_{3}-C_{4}+\beta B)+(1-x)(R_{2}+R_{3}-C_{4})$$
(5)$$u_{d}=x(R_{4}-C_{6}-\lambda D)+(1-x)(R_{4}-C_{6})$$
(6)$$\bar{u_{2}}=yu_{c}+(1-y)u_{d}$$

The replicator dynamics equation of the government can be calculated as follows:(7)$$F(x)=\frac{dx}{dt}=x\left(u_{a}-\bar{u_{1}}\right)=x\left(x-1\right)[y\left(\beta B+\lambda D\right)+\alpha A-\lambda D]$$

The replicator dynamics equation of the enterprises can be calculated as follows:(8)$$F(y)=\frac{dy}{dt}=y\left(u_{c}-\bar{u_{2}}\right)=y\left(1-y\right)[x\left(\beta B+\lambda D\right)+R_{3}-C_{4}-R_{4}+R_{2}+C_{6}]$$

According to Equation (7), if $y=\frac{\lambda D-\alpha A}{\beta B+\lambda D}$, then $\frac{dx}{dt}\equiv 0$, which means that the government has evolutionary stable strategy (ESS); if $y\neq \frac{\lambda D-\alpha A}{\beta B+\lambda D}$, then $x^{*}=0$ and $x^{*}=1$ are the two ESSs of the government, where $y>\frac{\lambda D-\alpha A}{\beta B+\lambda D}$, $x^{*}=0$ is ESS, $y<\frac{\lambda D-\alpha A}{\beta B+\lambda D}$, $x^{*}=1$ is ESS.

According to Equation (8), if $x=\frac{C_{4}+R_{4}-R_{2}-R_{3}-C_{6}}{\beta B+\lambda D}$, then $\frac{dy}{dt}\equiv 0$, which means that the enterprises have ESS; if $x\neq \frac{C_{4}+R_{4}-R_{2}-R_{3}-C_{6}}{\beta B+\lambda D}$, then $y^{*}=0$ and $y^{*}=1$ are the two ESSs of the enterprises, where $x>\frac{C_{4}+R_{4}-R_{2}-R_{3}-C_{6}}{\beta B+\lambda D}$, $y^{*}=1$ is ESS, $x<\frac{C_{4}+R_{4}-R_{2}-R_{3}-C_{6}}{\beta B+\lambda D}$, $y^{*}=0$ is ESS.

It is worth noting that this model is related to parameters λD, αA, $C_{4}$, $R_{4}$, $R_{2}$, $R_{3}$ and $C_{6}$. This paper discusses the differences in the cases $\frac{\lambda D-\alpha A}{\beta B+\lambda D}<0$ and $0<\frac{\lambda D-\alpha A}{\beta B+\lambda D}<1$, respectively.

(1)When $\frac{\lambda D-\alpha A}{\beta B+\lambda D}<0$, due to y∈[0,1], for any y, there exists $[y\left(\beta B+\lambda D\right)+\alpha A-\lambda D]>0$, at which point $x^{*}=0$ is ESS for the government.

If $R_{4}-C_{6}\ll R_{2}+R_{3}-C_{4}$, then $\frac{C_{4}+R_{4}-R_{2}-R_{3}-C_{6}}{\beta B+\lambda D}<0$, and x∈[0,1], so for any x, there exists $[x\left(\beta B+\lambda D\right)+R_{3}-C_{4}-R_{4}+R_{2}+C_{6}]>0$, at which point $y^{*}=1$ is ESS for the enterprise. That is, when the benefits of legal discharge are much more significant than illegal discharge, the negative regulation of the government and the legal discharge of enterprises are ESS.

If $R_{4}-C_{6}>R_{2}+R_{3}-C_{4}$, and $0<\frac{C_{4}+R_{4}-R_{2}-R_{3}-C_{6}}{\beta B+\lambda D}<1$, when $x>\frac{C_{4}+R_{4}-R_{2}-R_{3}-C_{6}}{\beta B+\lambda D}$, $y^{*}=1$ is ESS; when $x<\frac{C_{4}+R_{4}-R_{2}-R_{3}-C_{6}}{\beta B+\lambda D}$, $y^{*}=0$ is ESS. At this time, the negative regulation of the government and the illegal discharge of enterprises are ESS.

If $R_{4}-C_{6}>R_{2}+R_{3}-C_{4}$, and $\frac{C_{4}+R_{4}-R_{2}-R_{3}-C_{6}}{\beta B+\lambda D}>1$, but x∈[0,1], there exists $[x\left(\beta B+\lambda D\right)+R_{3}-C_{4}-R_{4}+R_{2}+C_{6}]<0$ for any x. At this time, $y^{*}=0$ is ESS for the enterprise. That is, when the benefits of illegal discharge are much more significant than legal discharge, the negative regulation of the government and the illegal discharge of enterprises are ESS.

When the revenue from the government’s fines λD are less than the costs of promoting ecological protection publicity αA, the enterprise’s behavior strategy depends on the relative size of the legal profits $R_{2}+R_{3}-C_{4}$ and the illegal profits $R_{4}-C_{6}$. When the profits from illegal discharge are less than the legal discharge, the ESS is government’s negative regulation and enterprises’ legal discharge. When the profits from illegal discharge are greater than legal discharge, the ESS is for both negative regulation and illegal discharge.

(2)When $0<\frac{\lambda D-\alpha A}{\beta B+\lambda D}<1$, $y>\frac{\lambda D-\alpha A}{\beta B+\lambda D}$, $x^{*}=0$ is ESS; when $y<\frac{\lambda D-\alpha A}{\beta B+\lambda D}$, $x^{*}=1$ is ESS.

If $R_{4}-C_{6}\ll R_{2}+R_{3}-C_{4}$, then $\frac{C_{4}+R_{4}-R_{2}-R_{3}-C_{6}}{\beta B+\lambda D}<0$, and x∈[0,1], so for any x, there exists $[x\left(\beta B+\lambda D\right)+R_{3}-C_{4}-R_{4}+R_{2}+C_{6}]>0$, at which point $y^{*}=1$ is ESS for the enterprise. That is, when the benefits of legal discharge are much more significant than illegal discharge, the negative regulation of the government and the legal discharge of enterprises is ESS.

If $R_{4}-C_{6}>R_{2}+R_{3}-C_{4}$, and $0<\frac{C_{4}+R_{4}-R_{2}-R_{3}-C_{6}}{\beta B+\lambda D}<1$, when $x>\frac{C_{4}+R_{4}-R_{2}-R_{3}-C_{6}}{\beta B+\lambda D}$, $y^{*}=1$ is ESS; when $x<\frac{C_{4}+R_{4}-R_{2}-R_{3}-C_{6}}{\beta B+\lambda D}$, $y^{*}=0$ is ESS. There is no ESS at this time.

If $R_{4}-C_{6}>R_{2}+R_{3}-C_{4}$, and $\frac{C_{4}+R_{4}-R_{2}-R_{3}-C_{6}}{\beta B+\lambda D}>1$, but x∈[0,1], there are $[x\left(\beta B+\lambda D\right)+R_{3}-C_{4}-R_{4}+R_{2}+C_{6}]<0$ for any x. At this time, $y^{*}=0$ is ESS for the enterprise. That is, when the benefits of illegal discharge are much more significant than legal discharge, the positive regulation of the government and the illegal discharge of enterprises are ESS.

When the revenue from the government’s fines λD are greater than the costs of promoting ecological protection publicity αA, the enterprise’s behavior strategy depends on the relative size of legal profits $R_{2}+R_{3}-C_{4}$ and the illegal profits $R_{4}-C_{6}$. When the profits from illegal discharge are less than the legal discharge, the ESS is for both the negative regulation and the legal discharge. When the profits from illegal discharge are greater than legal discharge, the ESS is for both the positive regulation and the illegal discharge.

In the two-party evolutionary game of government and enterprises, the government’s stability strategy mainly depends on the relative size of fines λD and the cost of promotion ecological protection publicity αA. Additionally, the enterprise’s behavior strategy depends on the relative size of the legal profits $R_{2}+R_{3}-C_{4}$ and the illegal profits $R_{4}-C_{6}$.

#### 3.1.2. The Equilibrium Analysis of the Public and Enterprises

This paper divides the public’s behavior into two categories in water environmental governance: public concern and public whistle-blowing. Public concern for the environment can effectively monitor the government’s adoption of appropriate management measures to reduce the harm caused by environmental pollution, thereby improving the urban environment effectively [33]. In addition, environmental protests in the form of letters, complaints, and other reports are also the expression of the public’s demands for the environment, which can be regarded as the effective participation of the public. This paper constructs the payoff matrix of the public and enterprises, as shown in Table 3.

The expected and average return of the public’s active participation and non-participation can be expressed as follows:(9)$$u_{e}=y\left(R_{5}+R_{6}-\varepsilon F-\sigma G\right)+(1-y)(-C_{5}-\varepsilon F-\sigma G)$$
(10)$$u_{f}=y\left(R_{5}\right)+(1-y)(-C_{5})$$
(11)$$\bar{u_{3}}=zu_{e}+(1-z)u_{f}$$

The expected and average return of the enterprises’ legal discharge and illegal discharge can be expressed as follows:(12)$$u_{g}=z(R_{2}+R_{3}-C_{4})+(1-z)(R_{2}+R_{3}-C_{4})$$
(13)$$u_{h}=z(R_{4}-C_{6}-C_{3})+(1-z)(R_{4}-C_{6})$$
(14)$$\bar{u_{4}}=yu_{g}+(1-y)u_{h}$$

The replicator dynamics equation of the public can be calculated as follows:(15)$$F(z)=\frac{dz}{dt}=z\left(u_{e}-\bar{u_{3}}\right)=z\left(1-z\right)[yR_{6}-\sigma G-\varepsilon F]$$

The replicator dynamics equation of the enterprises can be calculated as follows:(16)$$F(y)=\frac{dy}{dt}=y\left(u_{g}-\bar{u_{4}}\right)=y\left(1-y\right)[zC_{3}+R_{3}-C_{4}-R_{4}+R_{2}+C_{6}]$$

According to Equation (15), if $y=\frac{\sigma G+\varepsilon F}{R_{6}}$, then $\frac{dz}{dt}\equiv 0$, which means that the public has ESS; if $y\neq \frac{\sigma G+\varepsilon F}{R_{6}}$, then $z^{*}=0$ and $z^{*}=1$ are the two ESSs of the public, where $y>\frac{\sigma G+\varepsilon F}{R_{6}}$, $z^{*}=1$ is ESS, $y<\frac{\sigma G+\varepsilon F}{R_{6}}$, $z^{*}=0$ is ESS.

According to Equation (16), if $z=\frac{C_{4}+R_{4}-R_{2}-R_{3}-C_{6}}{C_{3}}$, then $\frac{dy}{dt}\equiv 0$, which means that the enterprises have ESS; if $z\neq \frac{C_{4}+R_{4}-R_{2}-R_{3}-C_{6}}{C_{3}}$, then $y^{*}=0$ and $y^{*}=1$ are the two ESSs of the enterprises, where $z>\frac{C_{4}+R_{4}-R_{2}-R_{3}-C_{6}}{C_{3}}$, $y^{*}=1$ is ESS, $z<\frac{C_{4}+R_{4}-R_{2}-R_{3}-C_{6}}{C_{3}}$, $y^{*}=0$ is ESS.

It is worth noting that this model is related to parameters σG, εF, $C_{4}$, $R_{4}$, $R_{2}$, $R_{3}$ and $C_{6}$. This paper discusses the differences in the cases $\frac{\sigma G+\varepsilon F}{R_{6}}>1$ and $0<\frac{\sigma G+\varepsilon F}{R_{6}}<1$, respectively.

(1)When $0<\frac{\sigma G+\varepsilon F}{R_{6}}<1$, $y>\frac{\sigma G+\varepsilon F}{R_{6}}$, $z^{*}=1$ is ESS; when $y<\frac{\sigma G+\varepsilon F}{R_{6}}$, $z^{*}=0$ is ESS.

If $R_{4}-C_{6}\ll R_{2}+R_{3}-C_{4}$, then $\frac{C_{4}+R_{4}-R_{2}-R_{3}-C_{6}}{C_{3}}<0$, and z∈[0,1], so for any z, there exists $[zC_{3}+R_{3}-C_{4}-R_{4}+R_{2}+R_{6}]>0$, at which point $y^{*}=1$ is ESS for the enterprise. That is, when the benefits of legal discharge are much more significant than illegal discharge, the active participation of the public and the legal discharge of enterprises are ESS.

If $R_{4}-C_{6}>R_{2}+R_{3}-C_{4}$, and $0<\frac{C_{4}+R_{4}-R_{2}-R_{3}-C_{6}}{C_{3}}<1$, when $z>\frac{C_{4}+R_{4}-R_{2}-R_{3}-C_{6}}{C_{3}}$, $y^{*}=1$ is ESS; when $z<\frac{C_{4}+R_{4}-R_{2}-R_{3}-C_{6}}{C_{3}}$, $y^{*}=0$ is ESS. At this time, there are two ESSs, one is non-participation and illegal discharge, and the other is active participation and legal discharge.

If $R_{4}-C_{6}>R_{2}+R_{3}-C_{4}$, and $\frac{C_{4}+R_{4}-R_{2}-R_{3}-C_{6}}{C_{3}}>1$, but x∈[0,1], there are $[zC_{3}+R_{3}-C_{4}-R_{4}+R_{2}+R_{6}]<0$ for any x. At this time, $y^{*}=0$ is ESS for the enterprise. That is, when the benefits of illegal discharge are much more significant than legal discharge, non-participation of the public and the illegal discharge of enterprises is ESS.

When the costs from the public’s participation σG+εF are less than the benefits $R_{6}$, the enterprise’s behavior strategy depends on the relative size of legal profits $R_{2}+R_{3}-C_{4}$ and the illegal profits $R_{4}-C_{6}$. It is worth noting that even if the benefits of legal discharge are smaller than illegal discharge, it is still possible for enterprises to choose legal discharge under the public participation.

(2)When $\frac{\sigma G+\varepsilon F}{R_{6}}>1$, due to y∈[0,1], for any z, there exists $[yR_{6}-\sigma G-\varepsilon F]<0$, at which point $z^{*}=0$ is ESS for the public.

If $R_{4}-C_{6}\ll R_{2}+R_{3}-C_{4}$, then $\frac{C_{4}+R_{4}-R_{2}-R_{3}-C_{6}}{C_{3}}<0$, and z∈[0,1], so for any z, there exists $[zC_{3}+R_{3}-C_{4}-R_{4}+R_{2}+R_{6}]>0$, at which point $y^{*}=1$ is ESS for the enterprise. Namely, when the benefits of legal discharge are much more significant than illegal discharge, non-participation of the public and the legal discharge of enterprises is ESS.

If $R_{4}-C_{6}>R_{2}+R_{3}-C_{4}$, and $0<\frac{C_{4}+R_{4}-R_{2}-R_{3}-C_{6}}{C_{3}}<1$, when $z>\frac{C_{4}+R_{4}-R_{2}-R_{3}-C_{6}}{C_{3}}$, $y^{*}=1$ is ESS; when $z<\frac{C_{4}+R_{4}-R_{2}-R_{3}-C_{6}}{C_{3}}$, $y^{*}=0$ is ESS. At this time, non-participation of the public and the illegal discharge of enterprises is ESS.

If $R_{4}-C_{6}>R_{2}+R_{3}-C_{4}$, and $\frac{C_{4}+R_{4}-R_{2}-R_{3}-C_{6}}{C_{3}}>1$, but z∈[0,1], there is $[zC_{3}+R_{3}-C_{4}-R_{4}+R_{2}+R_{6}]<0$ for any z. At this time, $y^{*}=0$ is ESS for the enterprise. That is, when the benefits of illegal discharge are much more significant than legal discharge, non-participation of the public and the illegal discharge of enterprises is ESS.

When the costs from the public participation σG+εF are higher than the benefits $R_{6}$, the enterprise’s behavior strategy depends on the relative size of legal profits $R_{2}+R_{3}-C_{4}$ and the illegal profits $R_{4}-C_{6}$. When the profits from illegal discharge are less than the legal discharge, the ESS is the non-participation and the legal discharge. When the profits from illegal discharge are greater than legal discharge, the ESS is the non-participation and the illegal discharge.

In the two-party evolutionary game of public and enterprises, the public’s stability strategy mainly depends on the relative size of costs σG+εF and benefits of public participation $R_{6}$. Additionally, the enterprise’s behavior strategy depends on the relative size of the legal profits $R_{2}+R_{3}-C_{4}$ and the illegal profits $R_{4}-C_{6}$. However, the above results also show that even if the benefits of legal discharge are smaller than illegal discharge, it is still possible for enterprises to choose legal discharge under public participation.

#### 3.1.3. The Equilibrium Analysis of Government, Enterprises and the Public

Water environment governance involves multiple stakeholders, and it has become a general trend to establish a co-governance system of government, enterprises, and the public. However, the interests and needs of all stakeholders are different, and the relationship is complex. Therefore, this paper constructs a tripartite evolutionary game model of government, enterprises, and the public and attempts to explore the optimal behavior strategies of each subject. The payoff matrix of government, enterprises, and the public is shown in Table 4.

The expected and average return of the government’s positive regulation and passive regulation can be expressed as follows:(17)$$\begin{array}{ll} u_{k}= & yz(R_{1}-\alpha A-\beta B)+y(1-z)(R_{1}-\alpha A-\beta B)\\ & +\;(1-y)z(\lambda D-\alpha A-C_{1}-C_{2})\\ & +\;(1-y)(1-z)(\lambda D-\alpha A-C_{1}) \end{array}$$
(18)$$u_{l}=yzR_{1}+y(1-z)R_{1}+(1-y)z(-C_{1}-C_{2})+(1-y)(1-z)(-C_{1})$$
(19)$$\bar{u_{5}}=xu_{k}+(1-x)u_{l}$$

The expected and average return of the enterprises’ legal discharge and illegal discharge can be expressed as follows:(20)$$\begin{array}{ll} u_{m}= & xz(R_{2}+R_{3}-C_{4}+\beta B)+x(1-z)(R_{2}+R_{3}-C_{4}+\beta B)\\ & +(1-x)z(R_{2}+R_{3}-C_{4})+(1-x)(1-z)(R_{2}+R_{3}-C_{4}) \end{array}$$
(21)$$\begin{array}{ll} u_{n}= & xz(R_{4}-C_{6}-C_{3}-\lambda D)+x(1-z)(R_{4}-C_{6}-\lambda D)\\ & +(1-x)z(R_{4}-C_{6}-C_{3})+(1-x)(1-z)(R_{4}-C_{6}) \end{array}$$
(22)$$\bar{u_{6}}=yu_{m}+(1-y)u_{n}$$

The expected and average return of the public’s active participation and non-participation can be expressed as follows:(23)$$\begin{array}{l} u_{p}=xy(R_{5}+R_{6}-\varepsilon F-\sigma G)+x(1-y)(-\varepsilon F-\sigma G-C_{5})\\ \;\;+(1-x)y(R_{5}+R_{6}-\varepsilon F-\sigma G)+(1-x)(1-y)(-\varepsilon F-\sigma G-C_{5}) \end{array}$$
(24)$$u_{q}=xyR_{5}+x(1-y)(-C_{5})+(1-x)yR_{5}+(1-x)(1-y)(-C_{5})$$
(25)$$\bar{u_{7}}=zu_{p}+(1-z)u_{q}$$

The replicator dynamics equation of the government can be calculated as follows:(26)$$F(x)=\frac{dx}{dt}=x\left(u_{k}-\bar{u_{5}}\right)=x\left(x-1\right)[\alpha A+y\beta B+(y-1)\lambda D]$$

The replicator dynamics equation of the enterprises can be calculated as follows:(27)$$F(y)=\frac{dy}{dt}=y\left(u_{m}-\bar{u_{6}}\right)=y\left(1-y\right)[x\left(\beta B+\lambda D\right)+R_{3}-C_{4}-R_{4}+R_{2}+C_{6}+zC_{3}]$$

The replicator dynamics equation of the public can be calculated as follows:(28)$$F(z)=\frac{dz}{dt}=z\left(u_{p}-\bar{u_{7}}\right)=z\left(1-z\right)[yR_{6}-\sigma G-\varepsilon F]$$

According to the conclusions of Ritzberger [34], the game model composed of government, enterprises, and the public only needs to discuss the equilibrium point at F(x)=0, F(y)=0 and F(z)=0, that is, the asymptotic stability of the equilibrium points at $E_{1}=(0,0,0)$, $E_{2}=(0,0,1)$, $E_{3}=(0,1,0)$, $E_{4}=(0,1,1)$, $E_{5}=(1,0,0)$, $E_{6}=(1,0,1)$, $E_{7}=(1,1,0)$ and $E_{8}=(1,1,1)$. The Jacobian matrix of the tripartite evolution system is:(29)$$J=\left(\begin{array}{ll} (2x-1)[\alpha A+y\beta B+(y-1)\lambda D]\;\;\;\;\;\;\;\;\;x(x-1)(\beta B+\lambda D)\;\;\;\;\;0\;\\ y(1-y)(\beta B+\lambda D)\;\;\;\;\;\;\;(1-2y)[x(\beta B+\lambda D) & \\ \;\;\;\;\;\;\;\;\;\;\;\;+R_{3}-C_{4}-R_{4}+R_{2}+C_{6}+zC_{3}]\;\;y(1-y)C_{3}\\ 0\;\;\;\;\;\;\;\;\;\;\;\;\;\;\;z(1-z)R_{6}\;\;\;(1-2z)(yR_{6}-\sigma G-\varepsilon F) \end{array}\right)$$

According to Lyapunov theory, a system’s asymptotic stability at the equilibrium point can be judged by the eigenvalues of its corresponding Jacobian matrix. When all the real parts of the eigenvalues are negative values, the equilibrium point is the ESS. When at least one of the real parts of the eigenvalues is a positive value, the equilibrium point is unstable [35]. Based on this, the eight equilibrium points were substituted into Equation (29) to obtain the eigenvalues at each equilibrium point, and the results are shown in Table 5.

Inference 1: when λD<αA and $R_{2}+R_{3}-C_{4}<R_{4}-C_{6}$, there exists a stable point E_1_ (0,0,0) in the replicative dynamic system.

It can be seen from inference 1 that the government’s behavior decision depends on the costs of promotion of ecological protection publicity and the size of the fines imposed on the enterprise. The enterprise’s behavior strategy depends on the relative size of the legal and the illegal profits. In the long-term, considering the costs of attention and supervision, the public is more inclined to adopt the decision of non-participation when the government is regulating negatively, and enterprises are discharging illegally.

Inference 2: when $R_{4}-C_{6}<R_{2}+R_{3}-C_{4}$ and $R_{6}<\sigma G+\varepsilon F$, there exists a stable point E_3_ (0,1,0) in the replicative dynamic system.

It can be seen from inference 2 that when the difference between the incomes and costs of legal discharge is more significant than that of illegal discharge, enterprises will adopt the strategy of legal discharge. When the additional benefits gained from public participation are less than the costs, the public lacks the incentive to participate actively. It thus tends to choose the behavior strategy of non-participation. Since legal discharge can also bring social benefits to the government, the government is more inclined to adopt the behavior strategy of negative regulations in this scenario, considering the costs of environmental advocacy and incentives.

Inference 3: when $R_{4}-C_{6}-C_{3}<R_{2}+R_{3}-C_{4}$ and $R_{6}>\sigma G+\varepsilon F$, there exists a stable point E_4_ (0,1,1) in the replicative dynamic system.

Inference 3 shows that enterprises will adopt legal discharge when the benefits of legal discharge are more excellent. When the additional benefits of public participation are more significant than the costs, the public chooses the behavior strategy of active participation. Considering that the government needs a specific cost to perform its responsibilities, it is more inclined to negative regulation in the case of legal discharge.

Inference 4: when αA<λD and $R_{2}+R_{3}-C_{4}+\beta B<R_{4}-C_{6}-\lambda D$, there exists a stable point E_5_ (1,0,0) in the replicative dynamic system.

It can be seen from inference 4 that when the costs of promotion ecological protection publicity are low, the government will take positive regulation. When the sum of the benefits of legal discharge and the government’s subsidies are less than the sum of the benefits of illegal discharge and fines, enterprises are more inclined to adopt illegal discharge. For the public, the increased costs of attention and monitoring reduce the public’s willingness to participate, leading them to choose the behavior strategy of non-participation.

### 3.2. Simulation Analysis

The above analysis clarifies the impact of various factors on behavior strategies from a theoretical perspective. Then, this paper simulates the above evolutionary game model based on Matlab through numerical and experimental methods to analyze the impact of different strengths of government action and public participation on enterprises’ emission behavior. In pollution control, the strategy choice of one side of the game subject will be affected by the proportion of other subjects’ decisions. Testing the evolution of group strategies by adjusting the initial value of one side can enhance the persuasiveness of the simulation results [36]. Initially, the probability of setting enterprise to choose different strategies is 0.3, 0.5, and 0.9, respectively, and the likelihood of the government and public choosing different strategies is x=0.5, z=0.5. To accurately reflect the evolutionary trajectory of the system, the time step is set to 0.1. The x, y, and z axes in Figure 1, Figure 2, Figure 3, Figure 4, Figure 5, Figure 6, Figure 7, Figure 8 and Figure 9 indicate the proportion of positive regulation, legal discharge, and active participation, respectively. Based on the above analysis, this paper discusses the influence of initial intention and execution intensity on behavior strategies.

#### 3.2.1. The Influence of Initial Intention on Behavioral Strategies

Scenario 1: When the costs of positive regulation and public participation are high, and the benefits of legal discharge are low, this paper sets *A* = 4, *B* = 3, *D* = 1, *R*_2_ = 8, *R*_3_ = 0.9, *R*_4_ = 10, *C*_3_ = 1, *C*_4_ = 8, *C*_6_ = 6, *R*_6_ = 0.9, *G* = 0.5, *F* = 0.7. With the increase in the probability of legal discharge, enterprises will evolve into a state of illegal discharge more slowly. The faster the government grows into a form of negative regulation, the slower the public’s willingness to participate will become non-participating after a period of growth. The final ESS is negative regulation, illegal discharge, and non-participation, with the evolutionary path of the system shown in Figure 1.

**Figure 1 ijerph-19-14732-f001:**
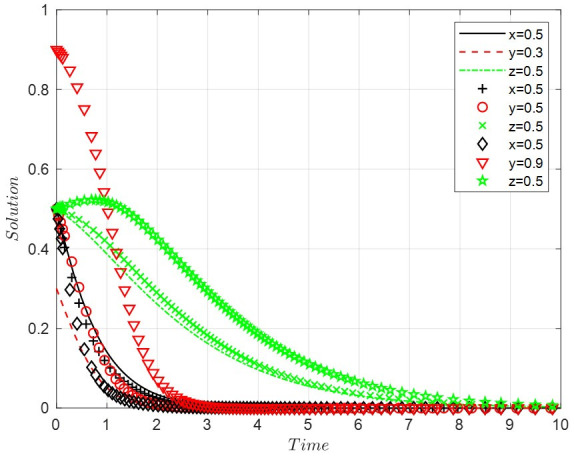
The evolutionary path of scenario 1.

Scenario 2: When the costs of positive regulation and public participation are high, and the benefits of legal discharge are high, this paper sets *A* = 3, *B* = 4, *D* = 1, *R*_2_ = 11, *R*_3_ = 0.9, *R*_4_ = 8, *C*_3_ = 1, *C*_4_ = 8, *C*_6_ = 6, *R*_6_ = 0.9, *G* = 2, *F* = 2. With the increase in the probability of legal discharge, the faster the enterprise evolves into the state of legal discharge, the faster the government develops into negative regulation, and the slower the public grows to the form of non-participation. The final ESS is negative regulation, legal discharge, and non-participation, with the evolutionary path of the system shown in Figure 2.

**Figure 2 ijerph-19-14732-f002:**
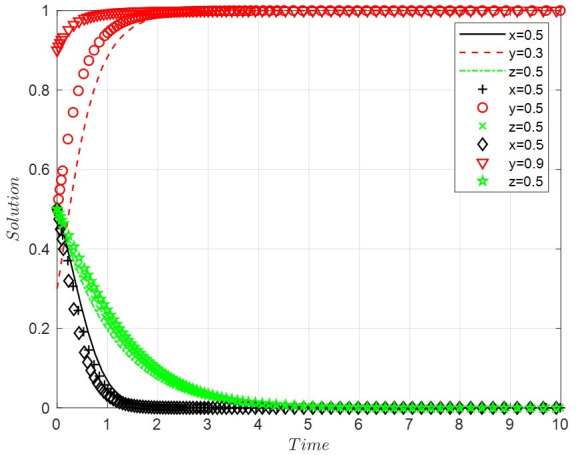
The evolutionary path of scenario 2.

Scenario 3: When the costs of positive regulation are high, the costs of public participation are low, and the benefits of legal discharge are high, this paper sets *A* = 3, *B* = 4, *D* = 1, *R*_2_ = 11, *R*_3_ = 0.9, *R*_4_ = 8, *C*_3_ = 1, *C*_4_ = 8, *C*_6_ = 6, *R*_6_ = 0.9, *G* = 0.5, *F* = 0.7. With the increase in the probability of legal discharge, the faster the enterprise evolves into legal discharge, the faster the government develops into negative regulation, and the quicker the public grows to the state of active participation. The final ESS is negative regulation, legal discharge, and active participation, with the evolutionary path of the system shown in Figure 3.

**Figure 3 ijerph-19-14732-f003:**
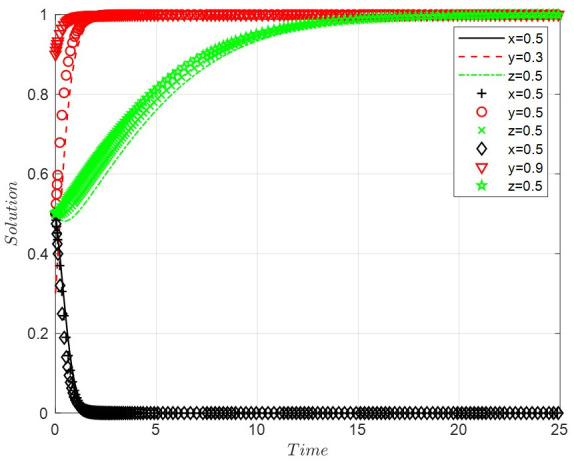
The evolutionary path of scenario 3.

Scenario 4: When the costs of positive regulation are low, the costs of public participation are high, and the benefits of legal discharge are low, this paper sets *A* = 1, *B* = 2, *D* = 4, *R*_2_ = 8, *R*_3_ = 0.9, *R*_4_ = 10, *C*_3_ = 1, *C*_4_ = 8, *C*_6_ = 6, *R*_6_ = 0.9, *G* = 0.5, *F* = 0.7. With the increase in the probability of legal discharge, the faster the enterprise evolves into the state of illegal discharge, the slower the government becomes into positive regulation. The public’s willingness to participate grows for a while and slowly tends towards non-participation. The final ESS is positive regulation, illegal discharge, and non-participation, with the evolutionary path of the system shown in Figure 4.

**Figure 4 ijerph-19-14732-f004:**
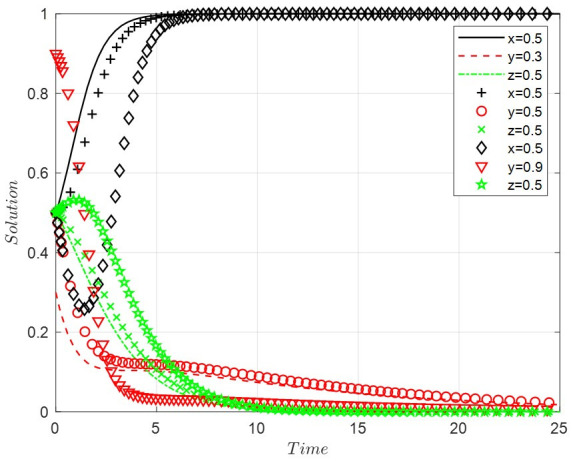
The evolutionary path of scenario 4.

Similarly, we can get the evolution path when the government and the public change their initial values and the other two game players have fixed initial values. Combined with the above simulations, the ESS of government, enterprises, and the public depend primarily on their costs and benefits. However, the convergence rate to the steady state may vary under different parameters. The ideal strategy is negative regulation, legal discharge, and non-participation. Therefore, this paper takes scenario two as the initial state to observe the effect of different types of government actions and public participation on enterprises.

#### 3.2.2. The Influence of Different Execution Intensity on Behavioral Strategies

(1)Simulation analysis under different ecological protection publicity

The simulation analysis is conducted when the intensity of the government’s ecological protection publicity α is 0.1, 0.5, and 0.9, respectively. The evolution path is shown in Figure 5. With the increase of α, the government, enterprises, and the public will choose negative regulation, legal discharge, and non-participation as their ESS. Therefore, the government’s ecological protection publicity promotes the legal act of enterprises, and highly intensive environmental protection publicity is conducive to more rapid legal discharge.

**Figure 5 ijerph-19-14732-f005:**
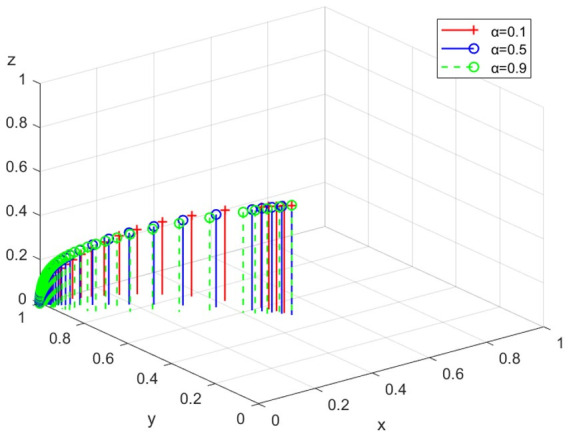
The path under different ecological protection publicity.

(2)Simulation analysis under different subsidies

The simulation analysis is conducted when the intensity of the government’s subsidy β is 0.1, 0.5, and 0.9, respectively. The evolution path is shown in Figure 6. With the increase of β, the government, enterprises, and the public will choose negative regulation, legal discharge, and non-participation as their ESS. Therefore, government subsidies encourage enterprises to choose legal discharge. When the subsidy standard is high, the evolution speed of enterprises towards legal discharge is faster.

**Figure 6 ijerph-19-14732-f006:**
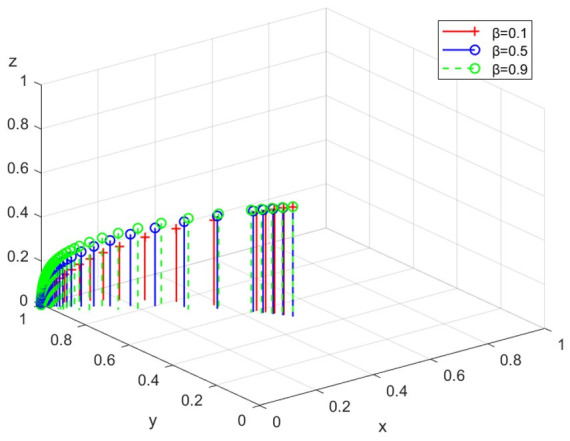
The path under different subsidies.

(3)Simulation analysis under different fines

The simulation analysis is conducted when the intensity of the government’s fine λ is 0.1, 0.5, and 0.9, respectively. The evolution path is shown in Figure 7. With the increase of λ, the evolution speed of enterprises’ choice of legal discharge continues to accelerate. Additionally, the ESS is negative regulation, legal discharge, and non-participation. Therefore, the fines imposed by the government, to a certain extent, restrict the behavior of illegal discharge.

**Figure 7 ijerph-19-14732-f007:**
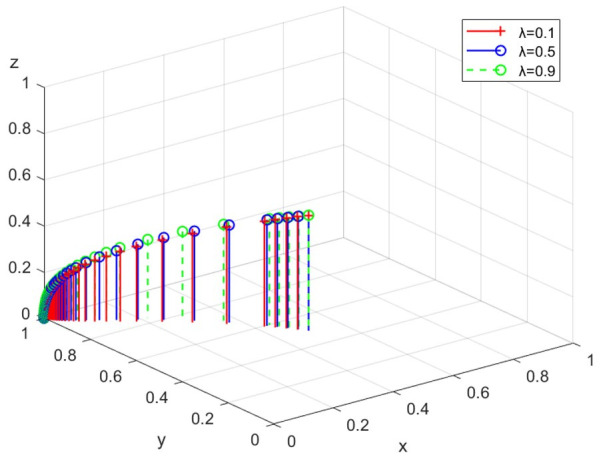
The path under different fines.

(4)Simulation analysis under different levels of public concern

The simulation analysis is conducted when the intensity of the public concern ε is 0.1, 0.5, and 0.9, respectively. The evolution path is shown in Figure 8. With the increase of ε, the evolution speed of enterprises towards legal discharge is faster. Public attention effectively pushes enterprises to choose legal discharge.

**Figure 8 ijerph-19-14732-f008:**
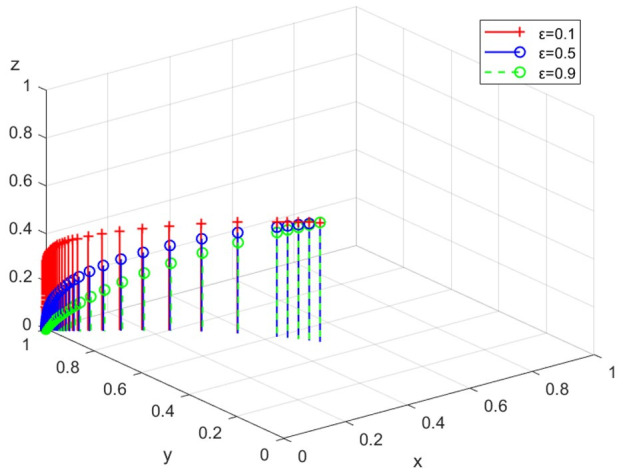
The path under different levels of public concern.

(5)Simulation analysis under different levels of public whistle-blowing

The simulation analysis is conducted when the intensity of public whistle-blowing σ is 0.1, 0.5, and 0.9, respectively. The evolution path is shown in Figure 9. With the increase of σ, the evolution speed of enterprises towards legal discharge is faster. The public tip-offs effectively encourage enterprises to choose legal discharge.

**Figure 9 ijerph-19-14732-f009:**
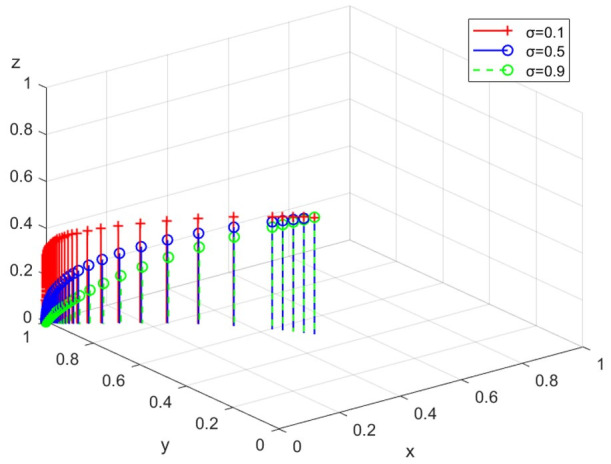
The path under different levels of public whistle-blowing.

## 4. Discussion

Based on the limited rationality of each stakeholder, this paper constructs the game models of government enterprise, public enterprise and government public enterprise, and explores the ESSs of each stakeholder in water environment governance. In the game equilibrium analysis of government, enterprise, and the public, the behavior of each stakeholder is related to its costs and benefits. There are commonalities and characteristics in the equilibrium analysis of the two-party and three-party evolutionary games. On the one hand, the steady-state conditions of government and the public are the same as those in the two-party evolutionary game models. On the other hand, the steady-state condition of enterprises needs to consider the impact on their reputation and image. Under effective public participation, enterprises may still choose to discharge legally, even if the benefits of legal discharge are smaller than illegal discharge.

The simulation results show that the government’s ecological protection publicity, subsidies, and fines have a positive role in promoting the legal discharge of enterprises. With the increase in execution intensity, enterprises tend to accelerate the evolution strategy of legal pollution discharge. As the authority of enterprise supervision, the government will restrain enterprises’ environmental pollution behavior. Therefore, we should give full play to the government’s leading role and establish and improve the government regulatory mechanism to effectively solve the problem of transboundary water pollution control. Water environment management is a persistent disease in public affairs, which needs to be solved through multi-party cooperation. The government should further improve the public participation mechanism in water pollution prevention so that the public has the right to know and participate.

The results show that public attention and whistle-blowing will affect the emission behavior of enterprises. Under effective public participation, it is still possible to choose legal discharge, even if the benefits of legal discharge are smaller than illegal discharge. With the gradual improvement of public awareness of environmental protection, the public has become an important force in environmental governance. As a non-governmental social force, the public is the direct recipient of environmental quality and can perceive the ecological situation in real-time. The active participation of the public can significantly improve the legal discharge of enterprises and urge the government to implement strict regulatory measures, thus helping to reduce the cost of environmental governance, which is economical. Therefore, in environmental management, the government should optimize the way of public participation, reduce the cost of public participation, and improve the enthusiasm for public participation.

From the results of game equilibrium analysis, the key to the legal discharge of enterprises lies in the relative size of costs and benefits. At the same time, the stability strategy of the enterprise is also affected by its reputation and image. As the main body of pollution prevention and control, the enterprise’s “short-termism” and the goal driven mainly by economic benefits quickly make it ignore the environmental benefits, thus resulting in illegal discharge. Therefore, enterprises should actively respond to the national call for energy conservation and emission reduction, change the traditional extensive production mode, and effectively control the increased pollutants.

## 5. Conclusions

Water environment governance involves many subjects, which is a typical complex system. This paper uses evolutionary game theory to analyze the influence of heterogeneous government behavior and public participation on the behavior of enterprises. The results show that in the game equilibrium analysis of the government, enterprises, and the public, the government’s and the public’s stability depend on the relative size of their respective costs and benefits. At the same time, the stability strategy of enterprises is also affected by their reputation and image. In addition, the government’s environmental publicity, subsidies, punishments, and the public’s attention and reports positively impact the behavior of enterprises’ legal discharge.

This paper explores achieving sustainable development through effective environmental policies to drive enterprises’ emission behavior. Based on the limitation of actual data, the parameters in this paper are only set subjectively by combining the theoretical analysis of the evolutionary game model with practical experience. The numerical simulation is conducted under conditions that have deviations from the actual situation. In future research, the theoretical model should be further combined with realistic data for empirical testing.

## Figures and Tables

**Table 1 ijerph-19-14732-t001:** Parameter symbols and their meanings.

Parameter	Meanings
αA	cost of promotion ecological protection publicity
βB	incentive costs of government
λD	government fines
εF	costs of public concern
σG	costs of public whistle-blowing
$R_{1}$	social benefits of government when enterprises discharge legally
$R_{2}$	economic benefits of legal discharge of enterprises
$R_{3}$	additional benefits of legal discharge of enterprises
$R_{4}$	economic benefits of illegal discharge of enterprises
$R_{5}$	benefits of the public when enterprises discharge legally
$R_{6}$	additional benefits from public participation
$C_{1}$	losses of government when enterprises discharge illegally
$C_{2}$	losses of government credibility caused by public whistle-blowing
$C_{3}$	losses of illegal discharge of enterprises
$C_{4}$	costs of legal discharge of enterprises
$C_{5}$	losses of the public when enterprises discharge illegally
$C_{6}$	costs of illegal discharge of enterprises

**Table 2 ijerph-19-14732-t002:** The payoff matrix of government and enterprises.

	Positive Regulation		Negative Regulation	
	Government	Enterprises	Government	Enterprises
legal discharge	$R_{1}-\alpha A-\beta B$	$R_{2}+R_{3}-C_{4}+\beta B$	$R_{1}$	$R_{2}+R_{3}-C_{4}$
illegal discharge	$\lambda D-\alpha A-C_{1}$	$R_{4}-C_{6}-\lambda D$	$-C_{1}$	$R_{4}-C_{6}$

**Table 3 ijerph-19-14732-t003:** The payoff matrix of the public and enterprises.

	Active Participation		Non-Participation	
	Public	Enterprises	Public	Enterprises
legal discharge	$R_{5}+R_{6}-\varepsilon F-\sigma G$	$R_{2}+R_{3}-C_{4}$	$R_{5}$	$R_{2}+R_{3}-C_{4}$
illegal discharge	$-C_{5}-\varepsilon F-\sigma G$	$R_{4}-C_{6}-C_{3}$	$-C_{5}$	$R_{4}-C_{6}$

**Table 4 ijerph-19-14732-t004:** The payoff matrix of government, enterprises and the public.

		Active Participation (*z*)	Non-Participation (1 − *z*)
positive regulation (*x*)	legal discharge (*y*)	$R_{1}-\alpha A-\beta B$	$R_{1}-\alpha A-\beta B$
		$R_{2}+R_{3}-C_{4}+\beta B$	$R_{2}+R_{3}-C_{4}+\beta B$
		$R_{5}+R_{6}-\varepsilon F-\sigma G$	$R_{5}$
	illegal discharge (1 − *y*)	$\lambda D-\alpha A-C_{1}-C_{2}$	$\lambda D-\alpha A-C_{1}$
		$R_{4}-C_{6}-C_{3}-\lambda D$	$\lambda D-\alpha A-C_{1}$
		$-\varepsilon F-\sigma G-C_{5}$	$-C_{5}$
negative regulation (1 − *x*)	legal discharge (*y*)	$R_{1}$	$R_{1}$
		$R_{2}+R_{3}-C_{4}$	$R_{2}+R_{3}-C_{4}$
		$R_{5}+R_{6}-\varepsilon F-\sigma G$	$R_{5}$
	illegal discharge (1 − *y*)	$-C_{1}-C_{2}$	$-C_{1}$
		$R_{4}-C_{6}-C_{3}$	$R_{4}-C_{6}$
		$-\varepsilon F-\sigma G-C_{5}$	$-C_{5}$

**Table 5 ijerph-19-14732-t005:** Stability analysis of equilibrium point.

Equilibrium Point	The Eigenvalues of Jacobian Matrix		Conclusion
	$\lambda_{1}$ $,\lambda_{2}$ $,\lambda_{3}$	Real Part	
E1 (0,0,0)	$-\alpha A+\lambda D,R_{3}-C_{4}-R_{4}+R_{2}+C_{6},-\sigma G-\varepsilon F$	(*,*,−)	ESS
E2 (0,0,1)	$-\alpha A+\lambda D,R_{3}-C_{4}-R_{4}+R_{2}+C_{6}+C_{3},\sigma G+\varepsilon F$	(*,*,+)	unstable point
E3 (0,1,0)	$-\alpha A-\beta B,-R_{3}+C_{4}+R_{4}-R_{2}-C_{6},R_{6}-\sigma G-\varepsilon F$	(−,*,*)	ESS
E4 (0,1,1)	$-\alpha A-\beta B,-R_{3}+C_{4}+R_{4}-R_{2}-C_{6}-C_{3},-R_{6}+\sigma G+\varepsilon F$	(−,*,*)	ESS
E5 (1,0,0)	$\alpha A-\lambda D,\beta B+\lambda D+R_{3}-C_{4}-R_{4}+R_{2}+C_{6},-\sigma G-\varepsilon F$	(*,*,−)	ESS
E6 (1,0,1)	$\alpha A-\lambda D,\beta B+\lambda D+R_{3}-C_{4}-R_{4}+R_{2}+C_{6}+C_{3},\sigma G+\varepsilon F$	(*,*,+)	unstable point
E7 (1,1,0)	$\alpha A+\beta B,-\beta B-\lambda D-R_{3}+C_{4}+R_{4}-R_{2}-C_{6},R_{6}-\sigma G-\varepsilon F$	(+,*,*)	unstable point
E8 (1,1,1)	$\alpha A+\beta B,-\beta B-\lambda D-R_{3}+C_{4}+R_{4}-R_{2}-C_{6}-C_{3},-R_{6}+\sigma G+\varepsilon F$	(+,*,*)	unstable point

Note: “+”, “−” and “*” respectively indicate that the real part symbol is positive, negative and unknown.

## Data Availability

Not applicable.

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
