# Peer review of "Research on Evolutionary Game of Water Environment Governance Behavior from the Perspective of Public Participation"

_ijerph, 2022, doi:10.3390/ijerph192214732_

Round 1

Reviewer 1 Report

The article is a very good read. Overall the article is well written, english language is almost perfect and is well structured. It provides a very interesting methodology and a set of results, based on evolutionary game theory that has been for the past year an active topic of research within colaborative planning. Conclusions and recommendations wrap up everything and bring the more practical application of the methodology to the real world, proving the importance of the whole study.

My only concern is the discussion. Basically the discussion is not really a discussion but rather the shortcomings of the project and after a several pages of a section 3., section 4. is just one paragraph.
However, along the article, in the 3. Results there is already a discussion and analysis of said results that is adequate. My advice would be to eliminate section 4. Discussion as it is serves no purpose and the shortcomings can be added as a subsection of 3.

Some small pointers:

LINE 68: “Many studies have demonstrated the importance of public participation in environmental governance from the aspects of participatory mechanism[15], participation mode[16], public attention[17] and so on”
There are in fact many studies and research regarding public participation in general and with focus on environmental governance. More references should be added, as saying “many” and just mentioning 3 is not enough.

LINE 79: I would suggest a rephrase in all the questions asked. For the reader it seems confusing. My suggestion would be to turn them into topics and not questions by improving their connection to what has been said in the last phrase and the “on the other hand” part.

LINE 116: The hypothesis are well presented but it still a lot of information for the reader. I would suggest adding a table resuming what has been said in the different hypothesis and explaining (in a more simplified manner) each variable.

Author Response

Dear Editor,

We appreciate editors and reviewers for all valuable comments and affirmation on the topic of the thesis, which play an essential role in improving the manuscript. According to the comments, we have revised the manuscript carefully. The responses towards the comments are as follows. We hope you will be satisfied with our answers.

Major Comment: The article is a very good read. Overall, the article is well written. English language is almost perfect and is well structured. It provides a very interesting methodology and a set of results, based on evolutionary game theory that has been for the past year an active topic of research within collaborative planning. Conclusions and recommendations wrap up everything and bring the more practical application of the methodology to the real world, proving the importance of the whole study.

Response: We express our sincere gratitude to the reviewer for your appreciation of the article.

A good ecological environment is the most inclusive people's livelihood and well-being. Since the reform and opening up, China's economy has achieved world-renowned achievements, with its GDP ranking second in the world, and its contribution to world economic growth has continuously exceeded 30%. But at the same time, with the gradual improvement of China's industrialization system and the gradual acceleration of the urbanization process, the contradiction between high energy consumption and severe economic structural transformation pressure and ecological environmental governance has seriously hindered high-quality economic development.

In recent years, the Chinese government has issued a series of policies and measures to strengthen water environment governance, which has effectively promoted the improvement of ecological water conditions. However, due to the complexity of water environment treatment, implementing policies has not yet achieved the expected effect. Besides, pollution control involves multiple stakeholders, and it is challenging to control enterprise pollution discharge effectively only by relying on government and market regulations. Based on the evolutionary game theory and stakeholder theory, this paper profoundly analyzes the impact of different stakeholders' games on water pollution prevention and control systems to provide references for water environment governance.

Comment#1: My only concern is the discussion. Basically, the discussion is not really a discussion but rather the shortcomings of the project. Along the article, in the 3. Results there is already a discussion and analysis of said results that is adequate. My advice would be to eliminate section 4. and the shortcomings can be added as a subsection of 3.

Response: Thank you for your helpful comments. We have rephrased the discussion and conclusion sections in the revised manuscript.

Comment#2: LINE 68: “Many studies have demonstrated the importance of public participation in environmental governance from the aspects of participatory mechanism [15], participation mode [16], public attention [17] and so on”. There are in fact many studies and research regarding public participation in general and with focus on environmental governance. More references should be added, as saying “many” and just mentioning 3 is not enough.

Response: Thank you for your helpful comments. We updated the references in the revised manuscript.

Revised: Many studies have demonstrated the importance of public participation in environmental governance from the aspects of participatory mechanism [19-20], participation mode [21-24], public attention [25-27], and so on.

[19] Chu, Z. P; Bian, C; Yang, J. How can public participation improve environmental governance in China? a policy simulation approach with multi-player evolutionary game. Environ. Impact. Assess. Rev. 2022, 95: 106782.

[20] Gera, W. Public participation in environmental governance in the Philippines: The challenge of consolidation in engaging the state. Land. Use. Pol. 2016, 52: 501-510.

[21] Fu, J. Y; Geng, Y. Y. Public participation, regulatory compliance and green development in China based on provincial panel data. J. Clean. Prod. 2019, 230: 1344-1353.

[22] Wu, J. N; Xu, M. M; Zhang P. The impacts of governmental performance assessment policy and citizen participation on improving environmental performance across Chinese provinces. J. Clean. Prod. 2018, 184: 227-238.

[23] Marco, G; Bo X. Air quality legislation and standards in the European Union: background, status and public participation. Adv. Clim. Chang. Res. 2013, 4(1): 50-59.

[24] Hasan, M. H; Nahiduzzaman, K. M; Aldosary, A. S. Public participation in EIA: A comparative study of the projects run by government and non-governmental organizations. Environ Impact Asses. 2018, 72: 12-24.

[25] Imane, E. I. O; Khaled, G; Jonathan, P; Andreas, Z. Public attention to environmental issues and stock market returns. Ecol. Econ. 2021, 180: 106836.

[26] Steffen, I. S; Eder, C. Public opinion in policy contexts. A comparative analysis of domestic energy policies and individual policy preferences in Europe. Int. Polit. Sci. Rev. 2020, 42(1).

[27] Gutsche, G; Ziegler, A. Which private investors are willing to pay for sustainable investments? Empirical evidence from stated choice experiments. J. Bank. Financ. 2019, 102: 193-214.

Comment#3: LINE 79: I would suggest a rephrase in all the questions asked. For the reader it seems confusing. My suggestion would be to turn them into topics and not questions by improving their connection to what has been said in the last phrase and the “on the other hand” part.

Response: We do appreciate your advice. This statement has been rewritten as advised.

Revised: On the other hand, many studies have discussed the influence of stakeholders on enterprise decision-making but have not yet considered the implementation strength of different stakeholders. However, multiple forms of government behavior and public participation often affect water environment governance. As a low-cost, efficient, and sustainable approach to water environment management, the public participation system is usually applied in practice in isolation from government governance. In addition, different forms of government behavior and public participation will affect the choice strategies of enterprises. Thus, this paper constructs an evolutionary game model of government, enterprises, and the public based on the bounded rationality assumption, considering the implementation strength of government and the public in water environment management. Then, through numerical simulation, this paper analyzes the impact of different intensities of government behavior and public participation on the choice of corporate behavior strategies, hoping to provide policy recommendations for building an environmental governance system dominated by the government, enterprises, and the public.

Comment#4: LINE 116: The hypotheses are well presented but it still a lot of information for the reader. I would suggest adding a table resuming what has been said in the different hypothesis and explaining (in a more simplified manner) each variable.

Response: We do appreciate your advice. We added a table to explain each parameter symbol and its meaning.

Revised: The above parameters and descriptions are shown in Table 1.

Table 1. Parameter symbols and their meanings

Parameter

meanings

αA

cost of promotion ecological protection publicity

βB

incentive costs of government

λD

government fines

εF

costs of public concern

σG

costs of public report

R1

social benefits of government when enterprises discharge legally

R2

economic benefits of legal discharge of enterprises

R3

additional benefits of legal discharge of enterprises

R4

economic benefits of illegal discharge of enterprises

R5

benefits of the public when enterprises discharge legally

R6

additional benefits from public participation

C1

losses of government when enterprises discharge illegally

C2

losses of government credibility caused by public report

C3

losses of illegal discharge of enterprises

C4

costs of legal discharge of enterprises

C5

losses of the public when enterprises discharge illegally

C6

costs of illegal discharge of enterprises

All relevant revisions have been marked by red font in the revised manuscript.

Thank you very much for your valuable comments and careful dedication. After these comments have been revised, the quality of the paper has been greatly improved. There may be some improper modifications in the article. Please criticize and correct again! In the process of revision, the author also revised the parts that could be improved. Please experts and editors to review! Thanks again!

Reviewer 2 Report

ID: ijerph-1966632-peer-review-v1

Title: Research on evolutionary game of water environment governance behavior from the perspective of public participation

The focus of the paper is on constructing the two-party evolutionary game model of government and enterprise, the two-party evolutionary game model of public and enterprise, and the three-party evolutionary game model of government, enterprise and the public. Authors have provided the analysis of the effects of different intensities of government actions and public participation on the behavior of enterprises.

In general, the paper requires a revision. I have some comments about organization and content of the paper. Please see the following comments:

-Figures have significant issue. First, the figures are black and white and so it is hard to analyse each data set. Please replace each figure with its color version to make it easier to difference between data sets. Second, and more importantly, Figure 1 is from a to d and Figure 2 is from a to e. I assume all Figure 1a, 1b, 1c and 1d should be located in the same place, not separated from each other. Then, it is easier for a reader to compare them. This is the common way of presenting figures in a paper. There is a same issue about Figure 2. If each of Figures 2a-e are independent, then number them as 2, 3, 4, 5 and 5. If not, place them in the same location (and page).

- The format of referencing papers in the body of the manuscript has some issues. The reference number should not be up script. For example, “serious in China[1-2]” should be replaced with “serious in China[1-2]” in Line 2 of the first page. Please correct the referencing format in the manuscript.

- Headings of sections are not informative as well. For example, Section 3 reveals the condition of formation of evolutionary stable strategies between different stakeholders through numerical simulation analysis. Its heading is “Result”. The single word “Result” neither if informative, nor is a good heading because the section is not about results. The proper heading should have words including “formation of evolutionary stable strategies between different stakeholders”.

- There are some cells in tables, e.g., Table 4, where words are broken. Equilib-rium, conclu-sion and unsta-ble are examples. Please never break a word in two lines when it is in a table. If you ask me, I believe that the format of tables requires a justification.

- I expected to see a more comprehensive literature review about the decision-making (not only game) of environment government and the public in term of costs and benefits. I suggest citing [a] Optimal decisions for competitive manufacturers under carbon tax and cap-and-trade policies. Computers & Industrial Engineering, vol.156, pp. 107244 [b] The impact of various carbon reduction policies on green flowshop scheduling, Applied Energy, vol.249, pp. 300-315

- As the final note, please merge Subsections 5.1 and 5.2 to have Section 5 as a single section with no subsections.

- The paper benefits from a proofread.

Author Response

Dear Editor,

We appreciate editors and reviewers for all valuable comments and affirmation on the topic of the thesis, which play an essential role in improving the manuscript. According to the comments, we have revised the manuscript carefully. The responses towards the comments are as follows. We hope you will be satisfied with our answers.

Major Comment: The focus of the paper is on constructing the two-party evolutionary game model of government and enterprise, the two-party evolutionary game model of public and enterprise, and the three-party evolutionary game model of government, enterprise and the public. Authors have provided the analysis of the effects of different intensities of government actions and public participation on the behavior of enterprises. In general, the paper requires a revision.

Response: Thank you for making time to review our manuscript. We are indeed very grateful and thankful for all your constructive comments. It really helps a lot in improving this manuscript.

Comment#1: Figures have significant issue. First, the figures are black and white and so it is hard to analyze each data set. Please replace each figure with its color version to make it easier to difference between data sets. Second, and more importantly, Figure 1 is from a to d and Figure 2 is from a to e. I assume all Figure 1a, 1b, 1c and 1d should be located in the same place, not separated from each other. Then, it is easier for a reader to compare them. This is the common way of presenting figures in a paper. There is a same issue about Figure 2. If each of Figures 2a-e are independent, then number them as 2, 3, 4, 5 and 5. If not, place them in the same location (and page).

Response: Thanks for your advice again, we have replaced each figure with its color version to make it easier to difference between data sets. In addition, we renumber each figure from Figure 1 to Figure 9.

Comment#2: The format of referencing papers in the body of the manuscript has some issues. The reference number should not be up script. For example, “serious in China [1-2]” should be replaced with “serious in China [1-2]” in Line 2 of the first page. Please correct the referencing format in the manuscript.

Response: We are sorry to make this mistake. And we have corrected the referencing format in the revised manuscript.

Comment#3: Headings of sections are not informative as well. For example, Section 3 reveals the condition of formation of evolutionary stable strategies between different stakeholders through numerical simulation analysis. Its heading is “Result”. The single word “Result” neither if informative, nor is a good heading because the section is not about results. The proper heading should have words including “formation of evolutionary stable strategies between different stakeholders”.

Response: We do appreciate your advice. Modification is made according to this advice. And we revised the headings of Section 3 "Results" to "Analysis of evolutionary stability strategies among different stakeholders".

Comment#4: There are some cells in tables, e.g., Table 4, where words are broken. Equilib-rium, conclu-sion and unsta-ble are examples. Please never break a word in two lines when it is in a table. If you ask me, I believe that the format of tables requires a justification.

Response: We are sorry to make this mistake. We have corrected all the tables in the revised manuscript.

Comment#5: I expected to see a more comprehensive literature review about the decision-making (not only game) of environment government and the public in term of costs and benefits. I suggest citing [a] Optimal decisions for competitive manufacturers under carbon tax and cap-and-trade policies. Computers & Industrial Engineering, vol.156, pp. 107244 [b] The impact of various carbon reduction policies on green flowshop scheduling, Applied Energy, vol.249, pp. 300-315

Response: Thank you for your comments, we have cited these papers (reference14 and reference18) in our revised manuscript.

Revised: (a) In recent years, with the continuous improvement of public environmental awareness, more and more people have begun to exercise the authority to supervise the behavior of the government and enterprises through government hotlines or media, and other channels [14].

(b) Sun [18] considers the public's environmental awareness and analyzes the enterprise's optimal strategy under different competitive behaviors and policy contexts.

(c) The problem of water environment governance not only stays at the technical level but also is a practical dilemma under different stakeholders' interest conflicts. Evolutionary game theory based on bounded rationality and group behavior analysis is increasingly used to reveal the behavior of complex subjects in environmental governance [14,18,28-29].

Comment#6: As the final note, please merge Subsections 5.1 and 5.2 to have Section 5 as a single section with no subsections.

Response: Thanks for your advice, we have rephrased the discussion and conclusion sections in the revised manuscript.

Comment#7: The paper benefits from a proofread.

Response: We do appreciate your advice. We have revised the manuscript carefully.

All relevant revisions have been marked by red font in the revised manuscript.

Thank you very much for your valuable comments and careful dedication. After these comments have been revised, the quality of the paper has been greatly improved. There may be some improper modifications in the article. Please criticize and correct again! In the process of revision, the author also revised the parts that could be improved. Please experts and editors to review! Thanks again!

Round 2

Reviewer 2 Report

All issues are resolved in the current version. I have read the paper quickly again to check the highlighted paragraphs. The game paper has already studied the problem on the sense of environmental requlations, proposed the method accurately, and significant extended the previous works.

My conclusion is that the paper can be accepted as it is.